# Research on the Coordinated Development of Population–Resources–Environment (PRE) Systems: An Empirical Analysis from Jiangsu Province, China

**DOI:** 10.3390/ijerph20010252

**Published:** 2022-12-23

**Authors:** Qian Zhao, Jianyuan Huang, Jiahao Yu, Xiao Du, Cong Li

**Affiliations:** 1Department of Sociology, Hohai University, Nanjing 211100, China; 2Population Research Institute, Hohai University, Nanjing 211100, China

**Keywords:** PRE systems, coordinated development, constraint analysis, spatial and temporal evolution, Jiangsu

## Abstract

As the population size and urbanization increase, the relationships within the population–resource–environment (PRE) systems are becoming tenser. Determining how to achieve the harmonious development of PRE systems is currently an important issue faced by society. This paper uses Jiangsu province in China as an example of constructing a coordinated development evaluation index system for PRE systems. Using the coefficient of the variation coordination method, we examined the comprehensive evaluation scores of PRE systems in Jiangsu province from 2000 to 2020 to explore its intersystem coordinated development evolution status and analyze its constraints. The results show that: (1) the overall evaluation scores of PRE systems in Jiangsu are on the rise; (2) the overall coordinated development degree of Jiangsu has undergone obvious changes in stage and is currently in the stage of coordinated development; (3) there are obvious spatial differences in the overall coordinated development degree of Jiangsu, and the overall trend is gradually changing from “low in the south and high in the north” to “high in the south and low in the north”; and (4) the population urbanization rate, environmental management effectiveness, and regional development imbalance, restrict the coordinated development of PRE systems in Jiangsu.

## 1. Introduction

Behind the development dividends, such as the global rise in population size and increases in the population’s quality of life and life expectancy, the development needs that humans have captured will also inevitably cause damage to resources and the environment [1]. As a prerequisite for sustainable social development and an important material basis for human survival and development [2], the increasingly prominent contradictions within population–resource–environment (PRE) systems have created new existential pressures for human development. Therefore, achieving the best possible balance among the population, resources, and environment through complex interactions and achieving the coordinated development of various elements in the social system has become a new global topic. As an emerging country, China’s expanding population size, rapid urbanization, and rapid economic progress have been among the key features of the country since its reform and opening up [3]. Population growth and development needs are worsening the ecological and environmental problems in the country, and the development of PRE systems is no longer balanced, which is a potential problem for socio-economic development. For this reason, the Chinese government included a green development plan in the 14th Five-Year Plan published in 2021 to “promote green development and harmonious coexistence between human beings and nature” with the aim of achieving the stable and coordinated development of the economy and society over a certain period within the constraints of the population, resources, and environment [4]. In such a context, it is particularly important to analyze the level and direction of the sustainable development of the PRE system by establishing a system index framework through a quantitative assessment to assist with the completion of the formulation of sustainable development policies in the region.

The coordinated development of the PRE system has long been a key concern of various countries. In recent years, with the gradual expansion of system-coordinated development research and other areas of study, scholars have begun to explore the coordinated development of multiple systems associated with the population, economy, resources, environment, etc., and to assess the degree of coordinated development between the population and resources and the environment by constructing a comprehensive evaluation index system or establishing a coupled coordination degree of multiple systems [5,6]. They have found that increased productivity can promote economic growth, but the transitional consumption of energy will increase pollutant emissions, which will intensify environmental damage [7,8]. Therefore, energy management is crucial for environmental security [9]. Increases in population density, urbanization development, and economic growth will hurt the environment while improving human capital utilization, and vigorous development and use of clean energy will help to reduce pollution, sustain long-term economic growth, and lead to improvements in environmental quality [10,11,12]. In terms of measurement methods, the coordinated development of the population and resources with that of the environment is usually analyzed by principal component analysis [13], system dynamics [14], data envelopment analysis [15], and the coupled coordination model [16,17]. For example, Chu et al. used coupled coordination models to measure the degree of sustainable development among the population, economy, and ecosystem in eastern Russia [18]. Lv et al. analyzed urbanization during economic transitions by measuring the factors influencing coordinated development among the population, land, and economy in Nanchang, China [19]. Chinese scholars, after evaluating the degree of coordinated development among the population, economy, and environment in study sites, concluded that some regions had entered the coordinated development stage but were still at a development bottleneck and had not progressed to a higher level [20]. Meanwhile, a double threshold effect of the population size and human capital was observed between the energy consumption intensity and carbon emissions, where the higher the population growth rate was, the higher the resource consumption was, so increasing the growth rate of human capital can alleviate the growth rate of the demand for resources and improve environmental quality [21].

After reviewing the corresponding literature, we concluded that previous research in this area had not only concentrated on economic issues in the establishment of an evaluation system but that there had also been a mix of the two system indicators when considering natural resources or ecological environment issues, which is not consistent with the concept of sustainable development. An example of this would be simply merging the two systems into the environment system [22], examining both the per capita resource occupancy (which has resource attributes) and the waste treatment capacity and wastewater treatment capacity (which are within the scope of environmental protection). In addition, current research prefers to select the extraction and utilization of non-renewable energy resources such as coal, oil, and natural gas as indicators when establishing the indicator system of resource systems [23,24]. While this orientation focuses on the finite nature of resources, it does not take into account the more important renewable resources (land, water, and climate resources) for the concept of sustainable development. At the same time, natural energy sources such as coal are not available in all regions for regional studies, and too much emphasis on the examination of a few energy sources may bring errors to systematic studies.

Based on the above deficiencies, it is necessary to redefine the connotation of the PRE system. As an interconnected complex system, it is important to focus not only on the sustainable development capacity of the system but also to consider the interaction mechanism between systems and the coordination capacity within subsystems when assessing its coordinated development degree. This study explains the coordinated development mechanism of the PRE system on the basis of the multisystem coordinated development structure proposed by Zeng [25] and Cao et al. [4]. The specific structure is shown in Figure 1.

The interactions among the population, resources, and environment are expressed as follows: (1) The population subsystem is the main body of the whole system, and the pursuit of the degree of the coordinated development of the PRE system is, in fact, to further guarantee the ability of human beings to survive and develop sustainably and to improve the well-being of people. Population subsystems can provide the necessary labor and technical conditions for their maintenance and governance, in addition to demanding the physical conditions needed from other systems. (2) The resource subsystem is the material basis for other systems. It not only provides the population subsystem with resources for development but also provides the material conditions needed for the improvement of the environmental system. At the same time, although the endless exploitation of resources by human beings puts pressure on the other two systems, it is also possible to increase the stock of resources while protecting the environment through the research and development of renewable resources and orderly exploitation. (3) The environmental subsystem is the spatial carrier of other systems, and there is also a relationship of conflict and coordination between it and other subsystems. On the one hand, people’s incorrect methods for resource extraction and utilization have changed the state and composition of the environment, and the improper treatment of factory waste discharge has caused serious damage to the ecological environment; on the other hand, people can publish active policies and measures to save the ecological environment.

In addition, specific studies have included biases through a tendency to examine along the temporal dimension while lacking integrated measurements for the evolutionary patterns of coordinated development, including the spatial dimension [26]. Therefore, this study aimed to test the evolutionary patterns associated with the coordinated development of regional PRE systems by applying geographic information systems (GIS) technology under the rational definition of resource and environmental system indicators, which fills a gap in existing studies.

As one of the fastest-growing provinces in China, Jiangsu Province has long faced relatively typical sustainable development problems. At the same time, its important development position in China (with a large population and highly developed economy) and the similarity of its internal development structure (southern, central, and northern Jiangsu) to the regional development structure of China as a whole (eastern, central, and western China) have been considered by Chinese scholars as study areas not only to clarify the development status of Jiangsu but also to refer to the national development situation to some extent. Therefore, a scientific assessment of the coordinated development of population, resource, and environmental systems in Jiangsu could be useful for exploring sustainable development paths in Jiangsu or China as a whole and in other emerging countries (regions) around the world. Therefore, in this study, we constructed a comprehensive evaluation system for the coordinated development of the population and resources and the environment in Jiangsu Province, China, as an example. This was used to study the coordinated development of PRE systems from 2000 to 2020 and to further analyze their constraints. It can act as a reference for decision-making to promote regional sustainable development. To our knowledge, this is the first study on the status of coordinated regional development using data from the seventh national population census of China.

The remaining sections of this paper are described below. Section 2 introduces the study area as well as the research methodology. Section 3 introduces the comprehensive evaluation scores of the population system, resource system, and environmental system in Jiangsu and the degree of coordinated development among the population, resources, and the environment in Jiangsu. Section 4 discusses the constraints associated with the coordinated development of the population, resources, and environment in Jiangsu. Section 5 summarizes the conclusions and proposes policy recommendations for reference.

## 2. Research Design and Methods

### 2.1. Research Area

Jiangsu Province is located in the middle of the eastern coastal region of China and has both a subtropical monsoon climate and a temperate monsoon climate. It is an important part of the Yangtze River Delta region. The province has a total area of 107,200,000 km^2^ and 13 municipalities, which can be divided into northern Jiangsu (Xuzhou, Lianyungang, Yancheng, Huaian, Suqian), central Jiangsu (Nantong, Taizhou, Yangzhou), and southern Jiangsu (Nanjing, Zhenjiang, Changzhou, Wuxi, Suzhou) according to their geographical locations ( see Figure 2). In 2020, the total population of Jiangsu Province was 84,748,000 at the time of the seventh national population census of China. The degree of urbanization was 73.44%, far exceeding the average level in China (63.84%). Meanwhile, in terms of natural resources, Jiangsu has a flat topography, dense water network, and land resources that are mainly in the plains, deep soil, and high fertility, so it is suitable for the development of farming operations. The monsoon climate is obvious, with significant precipitation. The Yangtze River crosses the east and west areas of the province, the Grand Canal runs through the north and south areas of the province, and the water transiting the province is abundant. Mineral resources are mainly manifested as “three more and three less”. In other words, there are more types of minerals but fewer reserves per capita, more small mines but fewer large mines, and more non-metallic mines but fewer metal mines. In addition, Jiangsu Province has a wide range of marine, land, biological, tourism, salt, chemical, oil, and gas resources.

With the expansion of the population of Jiangsu Province and the increase in the level of urbanization, problems regarding the decrease in per capita resources and increased environmental pressure have arisen. In 2020, the per capita arable land in Jiangsu Province was only 0.00057 km^2^, which is far below the national average of 0.00101 ${\mathrm{km}}^{2}$ and close to the alert line of 0.00053 km^2^ per capita established by the United Nations. There are insufficient local water resources in Jiangsu Province, with a per capita water resources occupancy of 641.3 m^3^, accounting for only 28.63% of the national level. In addition, the proportion of traditional industries with high rates of energy consumption and pollution in Jiangsu Province is still high, which aggravates industrial emissions. In the 2000–2020 period, industrial sulfur dioxide emissions in Jiangsu Province showed a decreasing spatial distribution from south to north, with an overall decreasing trend at the provincial level, but the consumption of sulfur-containing fuels still accounts for a high proportion of the total use by the country. Promoting the harmonious development of the population and resources with the environment is therefore necessary.

### 2.2. Construction of an Indicator System

The purpose of the study of coordinated development is to examine whether the population, resource, or environmental systems are within the carrying capacity of the other two systems and whether each system contributes to its development to the greatest extent possible while promoting the common development of the other systems. As the basis for conducting an assessment on the degree of coordinated development, the indicator system was used to measure the level of coordinated development in the system. Based on the index system detailed in the existing literature, this study comprehensively took into account the interactions within PRE systems. Based on the principles of objectivity, comprehensiveness, operability, and scientificity, an evaluation index system for the coordinated development of the PRE system was constructed.

As a general term for populations of a certain quantity and quality and with certain relationships in a specific geographical area, three indicators can be used to represent the population system: population size, population structure, and population quality [5,27]. Among these, the population size includes the natural population growth rate and the population density; the population structure includes the proportion of the population aged 65 and above, and the population urbanization rate; and the population quality includes the life expectancy per capita, and the number of invention patents owned per 10,000 people. The resource system is a general term for all objective elements that are linked to human socio-economic development, and which can be used for production and life. We used two indicators, resource endowment, and resource utilization, to evaluate the resource level of a region [28]. Resource endowment includes the water resource possession per capita and the land area owned per capita. Resource utilization includes the crop sown area per capita and the park green area per capita. Meanwhile, we evaluated the environmental level of the region using certain indicators (environmental pressure and environmental response) [16,29]. Environmental pressure mainly refers to the air quality and industrial pollution emission status, including the annual average concentration of respirable particulate matter and the industrial wastewater discharge compliance rate. The environmental response mainly refers to the treatment of the environment and ecological protection, including the comprehensive utilization rate of industrial solid waste and the urban municipal sewage treatment rate.

In summary, we constructed a PRE system evaluation index system that uses seven primary indicators and 14 secondary indicators (see Table 1) to measure the degree of the development of the population, resource, and environment systems in Jiangsu. The indicators in the system are directional in nature. Positive indicators promote the coordinated development of the system, where the larger the value of the indicator, the more favorable the coordinated development of the system. Negative indicators inhibit the coordinated development of the system, so the larger the value of the indicator, the less favorable the coordinated development of the system.

The data used in this research were taken from the “Jiangsu Statistical Yearbook (2000–2020)”, except for the annual average concentration of respirable particulate matter, the compliance rate of industrial wastewater discharge, and the comprehensive utilization rate of industrial solid waste, which were taken from the “Jiangsu Environment Bulletin (2000–2020)”. There were missing data for some years, so these values were projected according to historical trends using the linear interpolation method.

### 2.3. Research Methodology

#### 2.3.1. Evaluation Method

Due to the possible differences between the nature, scale, and measured levels of different indicators, the comprehensive evaluation process often requires the dimensionless processing of the indicator data. In this paper, the data standardization (Z-score) method was used to process the data in a dimensionless manner. The data normalization formula used was:(1)$$Z_{i}=\frac{x_{i}-{\bar{x}}_{i}}{s_{i}}$$
where ${\bar{x}}_{i}$ is the mean value of the *i*-th indicator, and $s_{i}$ is the standard deviation.

Considering that there are two types of indicators in the original index system, namely positive indicators with larger values and negative indicators with smaller values, there is no clear quantitative boundary between “good” and “bad” indicators. In this paper, the affiliation values of indicators in the index system were calculated using the half-ascending trapezoid and half-descending trapezoid fuzzy affiliation functions, respectively.

For the positive indicators, the half-ascending fuzzy affiliation function was used:(2)$${\bar{C}}_{i}=\frac{C_{i}-C_{min}}{C_{max}-C_{min}}$$

For the negative indicators, a half-decreasing fuzzy affiliation function was used:(3)$${\bar{C}}_{i}=\frac{C_{max}-C_{i}}{C_{max}-C_{min}}$$

The affiliation values of the indicators all fell in the interval of [0, 1], and $C_{max}$ and $C_{min}$ were taken as the upper and lower limit values of the indicators, respectively.

The entropy value method was used to assign weights to the indicators. Considering the calculation process used in the entropy value method, a more robust standardized formula was used in this paper, as follows:

Positive indicators:(4)$${\bar{C}}_{i}=\frac{C_{i}}{C_{max}+C_{min}}$$

Negative indicators:(5)$${\bar{C}}_{i}=1-\frac{C_{i}}{C_{max}+C_{min}}$$

The basic principle of the entropy method is that the greater the degree of difference between the index values of a certain indicator, the lower the information entropy, the more information provided by the indicator, and the greater the weight of the indicator. Conversely, the smaller the degree of difference between the values of a certain indicator, the greater the information entropy, the smaller the amount of information provided by the indicator, and the smaller the weight of the indicator. The calculation process is as follows:(6)$$e_{j}=-K\sum p_{ij}lnp_{ij}$$
(7)$$w_{j}=\frac{d_{j}}{\sum d_{j}}$$
where $p_{ij}$ is the standardized index value; *K* is a constant, generally taken as $K=\frac{1}{Ln\;m}$; *m* is the year; $e_{j}$ is the information entropy value of the index; $d_{j}$ is the redundancy of the index, calculated as $d_{j}=1-e_{j}$; and $w_{j}$ is the final weight of the evaluation index.

The final score of each system is:(8)$$U_{i}=\sum w_{ij}C_{ij}$$
where $U_{i}$ is the final score of the *i*-th system, $C_{ij}$ is the standardized value of the *j*-th index of the *i*-th system, and $w_{ij}$ is the weight corresponding to $C_{ij}$.

#### 2.3.2. Coupling Model and Coupling Coordination Degree Model

The coefficient of variation coordination, also known as discrete coefficient coordination, is a method used to find the coordination coefficient among multiple systems based on the concept and properties of the coefficient of variation and the coordination coefficient through mathematical and statistical analyses. It is based on observing the degree of variation within each observation.

Suppose that the composite scores of the three systems—population, resources, and the environment—are $U_{1},\;U_{2},$ and $U_{3}$, respectively. When $U_{1}=U_{2}=U_{3}$, the coordination coefficient is one; that is, the three systems are in the best coordination state. When the values of $U_{1},\;U_{2},$ and $U_{3}$ are not equal, the closer the values of the sample variables are, the greater the coordination degree is, and vice versa, the further apart the values of the sample variables are, the smaller the coordination degree is. The calculation formula is as follows:(9)$$C={\left[\frac{3\left(U_{1}U_{2}+U_{1}U_{3}+U_{2}U_{3}\right)}{{\left(U_{1}+U_{2}+U_{3}\right)}^{2}}\right]}^{3}$$

The coordination degree *C* is an important indicator that portrays the coordination between the population, resources, and the environment. It is important to measure the degree of coordinated development between the population, resources, and the environment. However, in some cases, it is difficult to reflect the overall function and development level of the population and resource environment. Therefore, in this study, we introduced the degree of coordination development to reflect the development of the population and resource environment. Compared with the coordination degree model, the coordination development degree has higher stability and a wider scope of application. The calculation method used for the coordinated development degree is as follows:(10)$$D=\sqrt{C*T}$$
(11)$$T=\alpha *U_{1}+\beta *U_{2}+\gamma *U_{3}$$
where *D* is the degree of coordinated development, *C* is the degree of coordination, *T* is the adjustment coefficient, which reflects the overall population and resource levels, and α, β, and γ are the weights.

#### 2.3.3. Model Construction of Driving Factors

According to the research on the driving factors of the coordinated development degree among regional systems in the existing literature [4,16,20,27], this paper takes the coordinated development degree as the explained variable. Additionally, six explanatory variables were selected based on the factors that might have an impact derived from the established studies. The specific variables are explained in Table 2.

After selecting the variables, this paper adopts multiple linear regression (MLR) to analyze the factors influencing the variation in the coupling degree of each system. In order to solve the estimation bias of the model as much as possible, the stepwise regression method is chosen to correct the model. The initial MLR mathematical equation is expressed as follows:(12)$$D_{t}=cons+\beta_{1}popd_{t}+\beta_{2}urb_{t}+\beta_{3}land_{t}+\beta_{4}csa_{t}+\beta_{5}rpm_{t}+\beta_{6}str_{t}+\varepsilon_{t}$$
where $D_{t}$ is the coupling coordination degree; *t* is the year; *cons* is a constant term; *popd* is the population density; *urb* indicates the population urbanization rate; *land* is the land area owned per capita; *csa* is the crop sown area per capita; *rpm* is the annual average concentration of respirable particulate matter; *str* is the urban municipal sewage treatment rate; $\beta_{1}$, $\beta_{2}$, …, and $\beta_{6}$ are the partial regression coefficients; and ε is a random perturbation term.

## 3. Research Results

### 3.1. Evaluation Results

The comprehensive evaluation scores of each system in Jiangsu Province are shown in Figure 3. Overall, from 2000 to 2020, the comprehensive evaluation scores of the population, resource, and environment systems in Jiangsu province showed increasing trends. For the resource system, two positive indicators, the completed housing area per capita and the land area per capita, had more positive impacts on the overall evaluation score than the crop sown area per capita and water resources per capita (both positive indicators). For the environmental system, compared with the annual average concentration of respirable particulate matter (negative indicator) and the comprehensive utilization rate of industrial solid waste (positive indicator), two positive indicators, the municipal wastewater treatment rate and industrial wastewater discharge compliance rate, were shown to have greater positive impacts on the overall evaluation score.

From 2000 to 2010, the score increased slowly from 0.05 to 0.27. After 2011, the score increased significantly, and after 2016, the growth rate exceeded that of the resource and environmental systems. In 2020, the score of the comprehensive evaluation of the population system reached 0.90, which is higher than those of the resource and environmental systems by 0.23 and 0.08, respectively. The continuous improvement of the comprehensive evaluation score of the population system was mainly influenced by positive indicators: the population urbanization rate, the number of invention patents per 10,000 people, and the continuous increase in life expectancy per capita. However, the impacts of negative indicators, population density, and the proportion of the population aged 65 and above were found to be on the rise. These negative effects were offset by the positive effects of positive indicators. After 2010, the rapid rise in the number of invention patents per 10,000 people contributed to a further increase in the positive effect of the comprehensive evaluation of the population system, making the comprehensive evaluation score of the population system exceed those of the resource and environment systems. From this trend, we can see that the core position of the population system is emerging, and it is clear that there is still room for improvement.

### 3.2. Estimation Results of the Degree of Coupling Coordination

Based on the comprehensive evaluation scores of the population, resource, and environment systems in Jiangsu, the changes in the degree of coordinated development of the population, resources, and environment in Jiangsu were further investigated using Equations (9)–(11).

Combining the stage characteristics of the rate of change in the coordination degree of the population, resources, and environment in Jiangsu and the results obtained from the GIS analysis software ArcGIS natural breakpoint method division, the coordination development degree was divided into six levels, as shown in Table 3.

According to the coordination degree classification criteria, the phase change characteristics of the overall coordination development degree of Jiangsu became more obvious during this period (Figure 4).

In the years from 2000 to 2004, the coupling coordination level was on the verge of the disorder stage. All population, resource, and environment systems developed slowly during this period, and the degree of development for the population system was much lower than those of the resource and environment systems. In terms of scores, the difference between the comprehensive evaluation scores of the population system and the resource system during this period ranged from 0.19 to 0.28, and the difference in comparison to the environment system ranged from 0.48 to 0.57. The significant development differences and convergence of the three systems led to fluctuations in the overall coordinated development degree in the lower-level range (0.26 to 0.32).

In the years from 2005 to 2010, the coupling coordination level was in the basic coupling coordination stage. In this stage, the resource and environmental systems were all developing steadily, the development speed of the population system began to accelerate, and the development gaps among the population, resource, and environmental systems continued to narrow. The gap between the comprehensive evaluation score of the population system and the resource system decreased from 0.28 to 0.21, and that of the environment system decreased from 0.57 to 0.44. The narrowing of the development gap between the systems caused the overall coordinated development degree to increase from 0.34 in 2005 to 0.50 in 2010.

In the years 2011 to 2013, the coupling coordination level was in the moderate coupling coordination stage. In 2013, the overall evaluation score of the population system exceeded that of the resource system and was only 0.04 lower than that of the environment system. The relatively low development rates of the resource and environment systems began to limit the increase in the overall coordinated development degree, and the overall coordinated development degree of the system at this stage only increased from 0.58 in 2011 to 0.64 in 2013.

In the years from 2014 to 2020, the coupling coordination level was in the coupling coordination and high-level coupling coordination stages. At this point, the development degree of the population system continued to increase, but the relatively slow development of the resource and environment systems meant that the overall coordinated development degree only increased from 0.67 in 2014 to 0.73 in 2020.

The coordinated development grades of the population and resources and the environment in each city of Jiangsu Province are shown in Table 4 and Figure 5. Overall, the coordinated development of the population, resource, and environmental systems in Jiangsu Province improved to a certain extent during 2000–2015, but the rate of change in the coordinated development of the region was not consistent. From the perspective of the time dimension, from 2000 to 2010, the coordinated system development in Jiangsu province maintained the pattern of “high in the north and low in the south” in space. In 2000, the systems were on the verge of a disorder stage in southern Jiangsu, central Jiangsu, and Xuzhou in northern Jiangsu, while the other regions in northern Jiangsu were at the stage of basic coordination. By 2005, the system status in central Suzhou was the first to change from the verge of disorder to basic coordination. This was mainly due to the increase in its per capita housing construction completion area, which led to a significant improvement in the comprehensive evaluation score of the resource system and promoted the matching of the development process with the population and environmental systems. By 2010, Nanjing, Zhenjiang, and Changzhou in southern Jiangsu had also attained the rank of basic coordination. The main reason for this was that the industrial layout of Jiangsu Province had been adjusted, and the industries that were originally distributed in the central and southern regions of Jiangsu Province were more polluting to the environment and had been gradually moved to the northern region of Jiangsu Province.

By 2015, the hierarchical pattern of coordinated system development in Jiangsu Province had changed more obviously. The overall trend changed from “low in the south to high in the north” to “high in the south to low in the north”. The South Jiangsu region had a good degree of coordinated system development, including in Nanjing and Zhenjiang, followed by Suzhou, Wuxi, and Changzhou (moderate coordination). All regions in Central Jiangsu also belonged to the moderate coordination stage. By contrast, the northern part of Jiangsu was still in a basic coordinated state and had not changed significantly compared with the previous decade. By 2020, Nanjing and Suzhou had reached the highly coordinated stage. The changes in the coordinated development of the Jiangsu system mainly stemmed from two aspects: on the one hand, there are differences in the level of education within the province of Jiangsu. The population innovation capacity in southern Jiangsu has increased substantially, as shown by the increasing number of invention patents per 10,000 people. This has accelerated the development of the population system. The northern part of Jiangsu Province is relatively backward in terms of education level and is less attractive for high-quality people, which makes its population innovation capacity relatively weak. On the other hand, the adjustment of the industrial structure layout and the enhancement of environmental management in Jiangsu province have had important influences on the change in hierarchical patterns for the coordinated development of the system. During this period, industries with the heavy environmental pollution in southern Jiangsu have been transferred to northern Jiangsu, which has started to focus on the development of high-tech industries. The negative impacts on resources and the environment are now greater, but thanks to the strengthening of environmental management in Jiangsu Province, the negative impact will not expand, and the coordinated development level of the system in northern Jiangsu Province can be maintained at the basic level.

### 3.3. Regression Results of the Driving Factors of Coupling Coordination Degree

The regression results in Table 5 show that although the overall fit of the initial model is good, there are still some variables that cannot pass the statistical test. It indicates that there is multicollinearity among the explanatory variables of the model, and some of the variables need to be eliminated appropriately; otherwise, it will lead to distortion in the estimation of the regression coefficients and affect the interpretation of the model results. In this paper, the stepwise regression method was used to eliminate the variables in the initial model and obtain the modified model estimation results, from which it can be seen that the overall model fitting effect has been improved. The estimation results of the modified model show that the urbanization rate has the most significant effect on the degree of coordinated development in terms of the PRE system in Jiangsu province, and the higher the urbanization rate, the higher the degree of coordinated development. Secondly, the crop sown area per capita also has a significant positive effect on the degree of coordinated development, which in a sense, can indicate that it is feasible to promote the coordinated development of the regional PRE system through the rational use of land. In addition, the annual average concentration of respirable particulate matter was also found to have a significant effect on the coordinated development of the region, which proves that the improvement in the coordinated development degree of the region is inseparable from the sustainable management of the environment.

## 4. Discussion

There is still much room for improving the coordinated development of the population, resources, and environment in Jiangsu. From the drivers revealed by the regression results, Jiangsu still needs to further increase the urbanization rate of its population while rationalizing the allocation of land and other resources, improving the efficiency of environmental management, and accelerating the construction of resource-saving and environmentally friendly cities. This finding is similar to the drivers derived by Kan et al. on the coordinated development of new urbanization and ecological environments [31], as well as to the drivers of the coupled coordination of systems by Cao et al. [4]. At the same time, following our comprehensive analysis of the degree of coordinated development among PRE systems in Jiangsu and their driving factors, we believe that the deep-seated constraints that may affect their coordinated development in the future are as follows.

### 4.1. An Increase in the Population Urbanization Rate Increases Pressure on the Resource and Environment

The process of population urbanization consumes large amounts of energy, mineral resources, and other materials, leading to the discharge of large amounts of polluting substances during the smelting process. This can cause problems such as acid rain, haze, and marine pollution [32]. The overall increase in the annual average concentration of nitrogen dioxide in Jiangsu from 2000 to 2020 is inextricably linked to factors such as the high urban electricity demand and the high automobile use due to the increased population density in urban areas of Jiangsu. However, resources are finite, and over-exploitation will disrupt the ecological balance through forest reduction, soil fertility loss, topsoil loss, desertification, and surface water pollution, which, in turn, will put great pressure on the environment. At the same time, the proportion of elderly people in the population is gradually increasing through the process of population urbanization, causing a relative decline in the age-appropriate labor force and an increasing number of people who need to be burdened per unit of the labor force. To a certain extent, this causes an excessive social burden on the employed population, which, in turn, leads to the rise of labor costs for enterprises, restricts the vitality of economic and social development, and ultimately hinders the comprehensive utilization of natural resources, such as water resources, arable land resources, and mineral resources [33].

### 4.2. Resource Use Efficiency Still Needs to Be Improved

Although *the* increasing rate of *population urbanization* can reach *the* level of coordinated system development by promoting the sharing of urban infrastructure and reducing the construction of duplicate facilities, it can also cause problems such as land occupation while expanding the scale of construction land [34]. In particular, the increasing rate of population urbanization has been accompanied by a continuous decrease in resources per capita, and the large population inflow and high natural population growth rates have led to increases in the population size and pressure on the supply of water, arable land, and other resources [35]. Second, in terms of the unreasonable application of chemical fertilizers and the low utilization rate of organic fertilizer resources, the excessive and blind application of fertilizers not only increases agricultural production costs and waste resources but also causes arable land slabbing and soil acidification and reduces the efficiency of arable land utilization. However, the improvement of arable land also has a positive effect on the coordinated development of the system. However, the reduction in arable land efficiency may lead to the blind expansion of arable land through deforestation, which in turn, has a negative impact on the environmental and resource subsystems [36]. In addition, the acceleration of urbanization and construction will inevitably lead to a reduction in the total amount of available water resources. In 2020, the size of the water resources in Jiangsu was only 641.3 $m^{3}\;\mathrm{per}\;\mathrm{capita}.$ Due to years of over-exploitation of groundwater, the water level continues to decline, and there is a contradiction between the supply and demand of water resources.

### 4.3. Avoid Failure to Carry Out Timely Environmental Management after Environmental Pollution

Jiangsu has a more typical “pollute first, treat later” industrial development characteristics, and environmental pollution problems are prominent. Jiangsu’s emissions of major pollutants are considerable, of which industrial emissions occupy a prime position. In 2018, there were 306,600 tons of sulfur dioxide emissions, of which industrial sulfur dioxide emissions accounted for 86.13%, and there were 332,800 tons of smoke (dust) emissions, of which industrial smoke (dust) emissions accounted for 87.05%. A large number of harmful gases present in industrial emissions can pose a serious threat to humans as well as to soil and water sources, which is detrimental to population development [37,38]. Studies have pointed out that long-term exposure to air pollution can cause cardiovascular and chronic respiratory diseases, immune system damage, and lung cancer, which can not only endanger the health of the population but even cause premature deaths [39]. Exposure of women to ambient air pollution during pregnancy can result in a significant decrease in fetal birth weight [40]. In addition, the improper disposal of solid wastes, such as straw, plastics, solid household waste, and dry organic fertilizers, can threaten the agroecosystem, affecting soil permeability, water, and nutrient transport, thus, inhibiting crop growth and reducing land productivity. Although energy conservation and emissions reduction have been vigorously promoted in Jiangsu in recent years, public awareness of energy conservation and environmental protection has been increasing, and the environmental quality has improved. Significant human, material, and financial resources are required to treat environmental pollution, and the effectiveness of treatment cannot be maintained for a long time [41]. Environmental management is not only a very long process but is also expensive, and the cost of environmental management is still increasing. Determining how to balance industrial development and environmental protection will require long-term focused attention.

### 4.4. The Imbalance in Regional Development Is More Obvious

An imbalance in regional development restricts the coordinated development of the overall system [41]. As shown in Figure 5, the degree of coordinated development for Jiangsu municipalities also shows significant regional differences. We believe that the following factors may be the main reasons for the formation of this phenomenon. First, in terms of educational development, the foundation of educational development is unbalanced across Jiangsu due to factors such as the level of economic development and urbanization. The economically developed region of southern Jiangsu has a relatively high investment in education, and the development of teachers and the education security capacity is greater than in central and northern Jiangsu, leading to the uneven development of the population quality in Jiangsu. This, in turn, hinders the coordinated development of the overall system. Second, in terms of matching the total amount of regional resources with the population, central and northern Jiangsu have rich non-metallic mineral resources, vast territories, and much greater land areas than southern Jiangsu, but the difference in the level of economic development between central and northern Jiangsu and southern Jiangsu has caused uneven distribution for the population in Jiangsu, meaning that the smaller resident population in northern Jiangsu has obtained more resources, while the larger resident population in southern Jiangsu has fewer resources. The unbalanced distribution of resources has had a negative impact on the overall coordinated development of Jiangsu. In addition, from the perspective of environmental protection and governance, due to the disparity in economic and social development levels, southern Jiangsu has a more adequate financial guarantee and technical support in environmental governance, and the effect of environmental governance is higher than that of other regions. By contrast, central Jiangsu and northern Jiangsu have invested less in human and material resources and technology. Therefore, the imbalance in regional development in Jiangsu is also an important factor that limits the overall coordinated development of the population, resources, and environment.

## 5. Conclusions and Policy Implications

### 5.1. Conclusions

This research paper took Jiangsu province as the research area and constructed an evaluation index system for the coordinated development of the population, resources, and environment. We used the coefficient of variation principle to construct a model for measuring the coordinated development degree and explored the coordinated development degree of the population, resources, and environment in Jiangsu province from 2000 to 2020 and its constraints.

First, the overall evaluation scores of the population, resource, and environment systems in Jiangsu Province showed an upward trend. In 2000–2020, the resource and environment systems in Jiangsu Province increased from 0.24 and 0.53 to 0.67 and 0.82, respectively, and the core position of the population system gradually emerged, as its initial overall score was found to be much lower than those of the resource and environment systems.

Second, the coordinated development level of the population, resource, and environmental systems in Jiangsu Province has been increasing, but the regional differentiation is obvious, and the overall pattern is “high in the south and low in the north”. From 2000 to 2010, the coordinated development level of the population, resource, and environmental systems in Jiangsu Province was “low in the south and high in the north” from 2000 to 2010. The development pattern of the population, resource, and environment systems in Jiangsu Province was “low in the south and high in the north”. By 2015, it had gradually changed to “high in the south and low in the north”. The degree of coordinated development among the systems increased significantly, whereby the areas on the verge of disorder disappeared, and some areas entered the stage of high coordination.

Third, in terms of drivers, increases in the urbanization rate, land use efficiency, and the degree of environmental governance were shown to be significantly and positively related to coordinated development.

### 5.2. Policy Implications

Changes in each system’s internal index will affect the development status of the system as well as the overall coordinated development status of the system. To promote the coordinated development of the population, resources, and environment in Jiangsu Province, the following countermeasures are proposed for consideration.

(1)The reasonable gathering of the population in urban clusters and the citizenship of the agricultural transfer population should be promoted. The population should be concentrated in urban agglomerations, and the main purpose of urban agglomerations would be to promote the coordinated development of large, small, and medium-sized cities and small towns with an emphasis on promoting the co-location of the Nanjing metropolitan area and the Nanjing–Zhenjiang–Yangzhou metropolitan area and the integration of the Suzhou–Wuxi–Changzhou metropolitan area. While fully respecting the right of the agricultural transfer population to make their own choices, we should accelerate the citizenship of the agricultural transfer population while creating conditions for them to truly become integrated and enjoy urban security.(2)The government needs to effectively use the existing resources and develop new green energy sources. First, the government needs to strengthen the legislation and improve the laws and regulations for protecting natural resources to ensure the sustainable use of resources. Second, the government needs to strictly control the exploitation of natural resources and improve the comprehensive utilization rate of resources in all aspects. In addition, the government needs to strengthen the development of new energy, eliminate the backward production capacity, and cultivate and grow the new energy industry. Finally, the government needs to promote the process of carbon neutralization for carbon peaks, comprehensively promote clean production, accelerate the green development of agriculture, and vigorously promote the energy-saving renovation of existing buildings and municipal infrastructures in cities and towns.(3)The ecology and environment department needs to improve environmental protection mechanisms and increase environmental protection governance. First of all, the department needs to grasp the work of air governance while controlling pollution from emissions. Second, the department needs to promote green low-carbon cycle developments, adjust the layout, scale, and structure of industries that do not meet the positioning of ecological and environmental functions and build a green industrial chain system. Finally, the department needs to improve and perfect the environmental protection system, provide corresponding encouragement and support to production activities and consumption behaviors that actively promote environmental protection, and at the same time, strengthen the reform of environmental supervision and implement the supervision of the whole society.(4)It is necessary to strengthen regional cooperation and narrow regional differences. Jiangsu must continue to promote the sharing of resources and use the “enclave economy” model to promote regional integration and development. The capital and technology from southern Jiangsu Province and the raw materials and labor resources of northern Jiangsu Province should be jointly developed to realize the complementary transfer of industries between northern Jiangsu Province and southern Jiangsu Province. The aim is to realize the development of industrial linkages and to promote the coordinated development of the population, resources, and environment in Jiangsu Province.

### 5.3. Limitation

This study has certain limitations. Due to the limitation of the data, this study only analyzed the PRE system in Jiangsu, which lacks practical significance for comprehensive development and the implementation of regional coordination policies compared to the subjects in a national or regional consortium (e.g., Yangtze River Delta). In addition, since the topic of this paper is to explore the degree of coordinated development of the PRE system in Jiangsu province, the influence factors of sub-regions are not explored when using multiple linear regressions to analyze the influence factors. In the future, more research objects and influencing indicators should be incorporated from a more macroscopic perspective to further expand research on the coordinated development of the PRE system.

## Figures and Tables

**Figure 1 ijerph-20-00252-f001:**
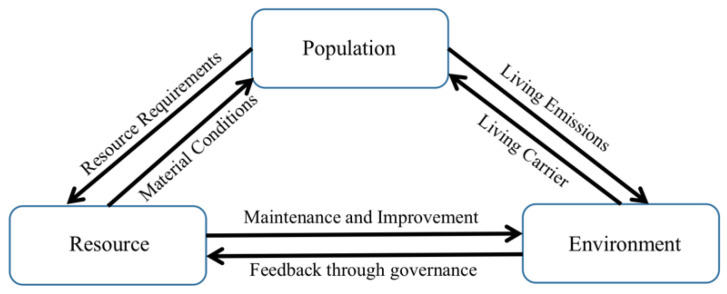
Interaction between PRE systems.

**Figure 2 ijerph-20-00252-f002:**
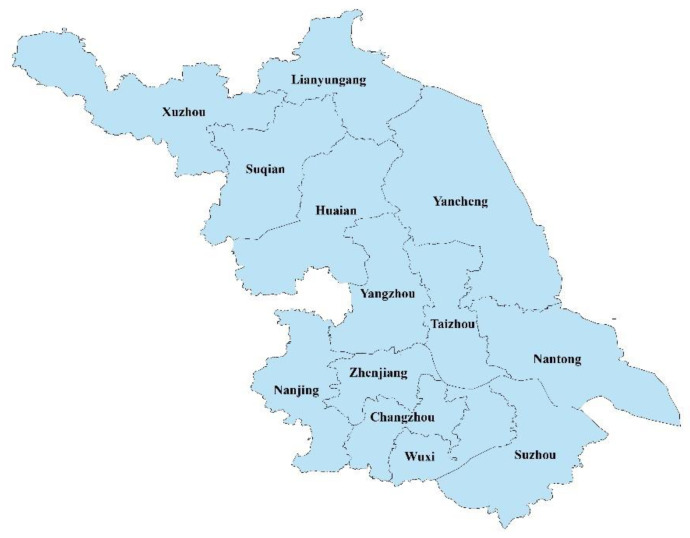
Map of Jiangsu Province.

**Figure 3 ijerph-20-00252-f003:**
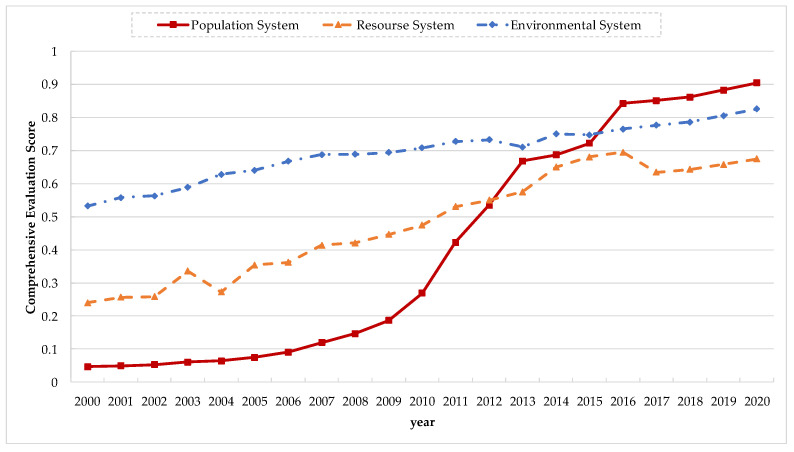
Time evolution of the comprehensive evaluation score of each system in Jiangsu.

**Figure 4 ijerph-20-00252-f004:**
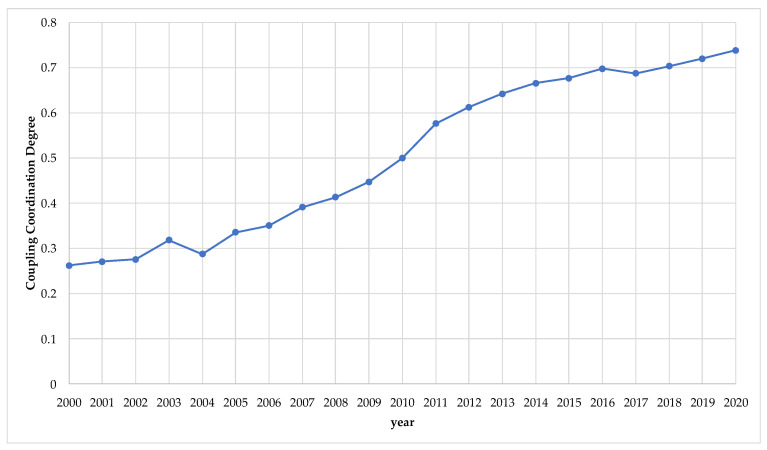
Time evolution of the degree of coordinated development of PRE systems in Jiangsu.

**Figure 5 ijerph-20-00252-f005:**
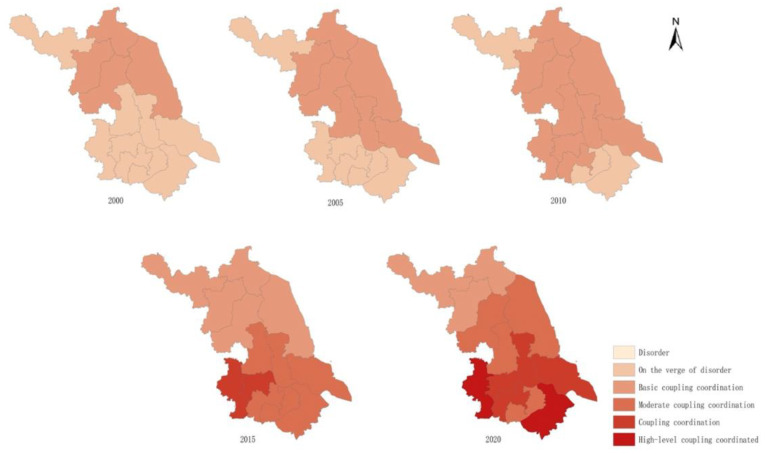
Spatial and temporal evolution of the degree of coordinated development of PRE systems in the cities of Jiangsu.

**Table 1 ijerph-20-00252-t001:** Coordinated development evaluation index system for PRE systems.

System Layer	Criterion Layer	Indicator Layer	Indicator Efficacy	References
Population System	Population Size	Natural population growth rate (‰)	−	[5,13,18,19]
		Population density (person/${\mathrm{km}}^{2}$)	−	[4,5,16,18,27,28]
	Population Structure	Percentage of population aged 65 and above (%)	−	[4,16,29,30]
		Population urbanization rate (%)	+	[4,16,27,28,29]
	Population quality	Life expectancy per capita (years)	+	[14,17,28]
		Number of invention patents per 10,000 people (pieces)	+	[19,27,28]
Resource System	Resource Endowment	Water possession per capita ($m^{3}$)	+	[4,16,18,19,22]
		Land area owned per capita ($m^{2}$)	+	[4,19,20,22,29]
	Resource Utilization	Crop sown area per capita (acres)	+	[4,19,22,29]
		Completed housing construction area per capita ($m^{2}$)	+	[22,23,24]
Environmental System	Environmental Pressure	Annual average concentration of respirable particulate matter (${\mathrm{mg}/m}^{3}$)	−	[4,16,27,28,29]
		Industrial wastewater discharge compliance rate (%)	+	[4,16,27,28,29]
	Environmental Response	Comprehensive utilization rate of industrial solid waste (%)	+	[4,19,20,27,29]
		Urban municipal sewage treatment rate (%)	+	[4,16,19,20,29,30]

**Table 2 ijerph-20-00252-t002:** Driving factors of coupling coordination degree.

Variable Type	Variable Name	Variable Symbol
Dependent variable	Coupling coordination degree	*D*
Independent variable	Population density	*popd*
	Population urbanization rate	*urb*
	Land area owned per capita	*land*
	Crop sown area per capita	*csa*
	Annual average concentration of respirable particulate matter	*rpm*
	Urban municipal sewage treatment rate	*str*

**Table 3 ijerph-20-00252-t003:** Classification of the coupling coordination degree.

Level	Coupling Coordination Degree	Coupling Coordination Status
I	0 ≤ D ≤ 0.24	Disorder
II	0.24 < D ≤ 0.32	On the verge of disorder
III	0.32 < D ≤ 0.50	Basic coupling coordination
IV	0.50 < D ≤ 0.65	Moderate coupling coordination
V	0.65 < D ≤ 0.70	Coupling coordination
VI	0.70 < D ≤ 1.00	High-level coupling coordinated

**Table 4 ijerph-20-00252-t004:** Coordinated development index and evaluation level of cities in Jiangsu.

Area	Index and Level	2000	2005	2010	2015	2020
Nanjing	Index	0.3014	0.3148	0.4651	0.6920	0.7424
	Level	II	II	III	V	VI
Suzhou	Index	0.2437	0.2651	0.2842	0.6242	0.7129
	Level	II	II	II	IV	VI
Wuxi	Index	0.2497	0.2419	0.3178	0.5811	0.6320
	Level	II	II	II	IV	IV
Changzhou	Index	0.2534	0.2706	0.3742	0.6191	0.6589
	Level	II	II	III	IV	V
Zhenjiang	Index	0.2692	0.2572	0.3605	0.6622	0.6694
	Level	II	II	III	V	V
Yangzhou	Index	0.3087	0.3643	0.4871	0.6119	0.6224
	Level	II	III	III	IV	IV
Nantong	Index	0.3097	0.3553	0.4610	0.6242	0.6608
	Level	II	III	III	IV	V
Taizhou	Index	0.2942	0.3726	0.4665	0.6240	0.6683
	Level	II	III	III	IV	V
Xuzhou	Index	0.2562	0.2826	0.3114	0.4293	0.4499
	Level	II	II	II	III	III
Lianyungang	Index	0.3415	0.3492	0.3426	0.4145	0.4263
	Level	III	III	III	III	III
Suqian	Index	0.3207	0.3525	0.3458	0.4052	0.4164
	Level	III	III	III	III	III
Huaian	Index	0.3593	0.3858	0.4204	0.4790	0.5486
	Level	III	III	III	III	IV
Yancheng	Index	0.3337	0.3866	0.4092	0.4811	0.5421
	Level	III	III	III	III	IV

**Table 5 ijerph-20-00252-t005:** Regression results.

Variable	Initial Model			Modified Model		
	Coefficient	*t* Statistic	*p* Value	Coefficient	*t* Statistic	*p* Value
*cons*	0.4867 ***	98.742	0.0000	0.4868 ***	105.555	0.0000
*popd*	0.0194	1.031	0.0002			
*urb*	0.1877 ***	5.123	0.3201	0.1673 ***	9.830	0.0000
*land*	−0.0447	−1.438	0.1724			
*csa*	0.0348	1.346	0.1996	0.0511 **	3.494	0.0030
*rpm*	0.0205	1.903	0.0778	0.0195 *	2.435	0.0269
*str*	−0.0327	−1.019	0.3260			
Adjust-R^2^	0.9876			0.9881		
F	267.2			414.5		

Note: *, **, and *** indicate that the variable is significant at the level of 10%, 5%, and 1%, respectively.

## Data Availability

Jiangsu Statistical Yearbook (2000–2020): http://tj.jiangsu.gov.cn/col/col4091/index.html (accessed on 18 June 2022); Jiangsu Environment Bulletin (2000–2020): http://sthjt.jiangsu.gov.cn/col/col83855/index.html (accessed on 18 June 2022).

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
