# Peer review of "Research on the Coordinated Development of Population–Resources–Environment (PRE) Systems: An Empirical Analysis from Jiangsu Province, China"

_ijerph, 2022, doi:10.3390/ijerph20010252_

Round 1

Reviewer 1 Report

Although the paper is the first study of the status of coordinated regional development using data from the seventh national population census of China.But, there are too many some studies, how to reflect the innovation of the paper needs authors to think.

 And other advice:

1.Construction of an Indicator System should add reference.

2.Research results should add analysis of driving force.

Author Response

Dear Reviewer:

We feel great thanks for your professional review work on our manuscript entitled “Research on the Coordinated Development of Population-Resources-Environment (PRE) Systems: An Empirical Analysis from Jiangsu Province, China”(ID: ijerph-2038453). Those comments are all valuable and very helpful for revising and improving our paper, as well as the important guiding significance to our research. We have studied the comments carefully and have made corrections which we hope meet with approval. Revised portions are marked in red on the resubmitted paper. The main corrections in the paper and the responses to the reviewer’s comments are as follows:

Response to the reviewer#1’s comments:

Comment 1: 

Although the paper is the first study of the status of coordinated regional development using data from the seventh national population census of China.But, there are too many some studies, how to reflect the innovation of the paper needs authors to think.

Response to comment 1: 

We thank the reviewer for the valuable comment. We think this is an excellent suggestion. As you pointed out, we did not emphasize the innovative nature of our article in the previously submitted manuscript, which was an oversight on our part. In our knowledge, this study differs from existing studies in that we have attempted to construct a relatively new structure of coordinated PRE development. Based on the review of existing studies, we believe that the previous PRE system architecture suffers from both an over-focus on economic issues and a mix of indicators in constructing resource and environmental systems.

First, we argue that too much focus on economic issues may lead to a misunderstanding of the relationship between the PRE system and the economy. The economic transformation process among population, resources and environment is a broad material transformation process. Population, resources and the environment have both natural and social material properties.Its development is not or not entirely the product of natural material transformation process in the narrow sense, nor is it the product of socio-economic process in the narrow sense. The economic linkage and economic process between population, resources and environment is a combination of material transformation in the broad sense and economic activity in the broad sense. It is a process of transformation of non-economic process into economic process, a process of transformation of externalities into internalities. Therefore, rushing to pursue the coordinated development of the economy and PRE system on the basis of not yet establishing the elements of coordinated development of population, resources and environment may make the whole system ambiguous. In fact, we find that in some studies it has already appeared that in the examination of the PREE system, not only the economic development elements are examined in the economic system, but also the economic elements are mixed in the resource and environmental system (e.g., the GDP conversion rate per unit of energy).

Second, there is a mix of resource and environmental system indicators in the existing literature. Some studies simply combine the two systems into one resource-environment system. Both the per capita resource possession (which has resource attributes) and the waste treatment capacity and wastewater treatment capacity (which are under the scope of environmental protection) are examined.

Combining the above factors, this study further attempts to establish a coordinated development mechanism for the PRE system based on the existing multi-system coordinated development structure, and carries out follow-up work based on this.

The corresponding new content is added in lines 81 to 125, page 2 to 3.

Comment 2:

Construction of an Indicator System should add reference.

Response to comment 2:

We thank the reviewer for pointing this out. We have added references to the index system in the manuscript.

The corresponding new content is added in lines 210 to 221, page 5 to 6, and table 1.

Comment 3: 

Research results should add analysis of driving force.

Response to comment 3:

We thank the reviewer for this comment and feel sorry for not analyzing our data in depth. To address this issue, we refer to some existing literature for an in-depth analysis of the data results. We use multiple linear regression to analyze the drivers of the changes in the coupling degree of each system. Also, to solve the estimation bias of the model as much as possible, the stepwise regression method is chosen to modify the model. After deriving the regression results of the model, we realigned the discussion section and the conclusion section.

The corresponding new content is added in lines 307 to 324, page 8 to 9; lines 451-472, page 13;lines 474 to 521, page 13 to 14;lines 602 to 604, page 16.

We tried our best to improve the manuscript and made some changes in the manuscript. And here we did not list the changes but marked them in red in the revised paper. We appreciate for reviewer’s warm work earnestly and hope that the correction will meet with approval. If there are any other modifications we could make, we would like very much to modify them and appreciate your help.

Once again, thank you very much for your comments and suggestions.

With best regards,

Authors

Reviewer 2 Report

This is an original and well-structured paper. However, there are still some issues to be solved before this manuscript can be published.

1.     The paper introduces the research background in a large amount of space. The author does not need to spend lots of effort to discuss it and should focus more on the research topic itself.

2.     In section 1, the paper points out that previous studies have confused the two systems of indicators, but have failed to provide a definition of the population-resource-environment system in this study.

3.     In 2.2, this paper does not specify which literature the index system is based.

4.     In 4.1, the author should pay attention to the analysis and discussion of the issue rather than the listing of data.

5.     The conclusions are not well presented. I do not think the results of the paper are sufficient to support conclusion 3. Is it scientific to draw conclusion 3 without empirical research and theoretical analysis? If so, can you explain them in discussion and draw more findings from the model?

6.     Limitations of the research should be introduced in the Conclusions.

Author Response

Dear Reviewer:

We feel great thanks for your professional review work on our manuscript entitled “Research on the Coordinated Development of Population-Resources-Environment (PRE) Systems: An Empirical Analysis from Jiangsu Province, China”(ID: ijerph-2038453). Those comments are all valuable and very helpful for revising and improving our paper, as well as the important guiding significance to our research. We have studied the comments carefully and have made corrections which we hope meet with approval. Revised portions are marked in red on the resubmitted paper. The main corrections in the paper and the responses to the reviewer’s comments are as follows:

Response to the reviewer#2’s comments:

Comment 1: 

The paper introduces the research background in a large amount of space. The author does not need to spend lots of effort to discuss it and should focus more on the research topic itself.

Response to comment 1: 

We thank the reviewer for the valuable comment. We think this is an excellent suggestion. We have further revised the section on research background based on your suggestions, reducing its non-essential elaboration and adding content that is more closely related to our research topic.

The corresponding new content is added in lines 28 to 96, page 1 to 2.

Comment 2:

In section 1, the paper points out that previous studies have confused the two systems of indicators, but have failed to provide a definition of the population-resource-environment system in this study.

Response to comment 2:

We thank the reviewer for pointing this out. With reference to the existing literature, we provide a definition of the population-resource-environment system in this study based on a comprehensive consideration of the sustainability of the PRE system, the interaction mechanisms and the internal coordination capabilities between the subsystems.

The corresponding new content is added in lines 97 to 125, page 3.

Comment 3: 

In 2.2, this paper does not specify which literature the index system is based.

Response to comment 3:

We thank the reviewer for pointing this out. We have added references to the index system in the manuscript.

The corresponding new content is added in lines 210 to 221, page 5 to 6, and table 1.

Comment 4:

In 4.1, the author should pay attention to the analysis and discussion of the issue rather than the listing of data.

Response to comment 4:

We thank the reviewer for this comment. Since we have added empirical research on the drivers of coordinated developmental change based on comment 5, the original discussion is no longer appropriate, so we have reorganized this section and avoided this issue you raised as much as possible.

The corresponding new content is added in line 474 to 521, page 13 to 14.

Comment 5:

The conclusions are not well presented. I do not think the results of the paper are sufficient to support conclusion 3. Is it scientific to draw conclusion 3 without empirical research and theoretical analysis? If so, can you explain them in discussion and draw more findings from the model?

Response to comment 5:

We thank the reviewer for this comment and feel sorry for not analyzing our data in depth. We thank the reviewer for this comment and apologize for reaching a conclusion without an in-depth analysis. As you said, it is not scientific to draw conclusion 3 without conducting empirical research. In fact, after we added the study of the drivers, we found that the actual results were not exactly in line with what we had concluded based only on the change in coordination. To address this issue, we refer to some existing literature for an in-depth analysis of the data results. We use multiple linear regression (MLR) to analyze the drivers of the changes in the coupling degree of each system. Also, to solve the estimation bias of the model as much as possible, the stepwise regression method is chosen to modify the model. After deriving the regression results of the model, we realigned the discussion section and the conclusion section.

The corresponding new content is added in lines 307 to 324, page 8 to 9; lines 451-472, page 13;lines 474 to 521, page 13 to 14;lines 602 to 604, page 16.

Comment 6:

Limitations of the research should be introduced in the Conclusions.

Response to comment 6:

We thank the reviewer for this reminder, and we have added a new section on the limitations of this study in section 5.3 of the manuscript.

The corresponding new content is added in lines 650 to 660, page 17.

We tried our best to improve the manuscript and made some changes in the manuscript. And here we did not list the changes but marked them in red in the revised paper. We appreciate for reviewer’s warm work earnestly and hope that the correction will meet with approval. If there are any other modifications we could make, we would like very much to modify them and appreciate your help.

Once again, thank you very much for your comments and suggestions.

With best regards,

Authors

Round 2

Reviewer 1 Report

All question had revised.The paper can accept in present form.

Reviewer 2 Report

All the issues are solved.